# Estimation of Time-Frequency Muscle Synergy in Wrist Movements

**DOI:** 10.3390/e24050707

**Published:** 2022-05-16

**Authors:** Ping Xie, Qingya Chang, Yuanyuan Zhang, Xiaojiao Dong, Jinxu Yu, Xiaoling Chen

**Affiliations:** Key Laboratory of Measurement Technology and Instrumentation of Hebei Province, Institute of Electric Engineering, Yanshan University, Qinhuangdao 066004, China; pingx@ysu.edu.cn (P.X.); qychang1122@163.com (Q.C.); dxjysdx1227@163.com (X.D.); yujinxu@stumail.ysu.edu.cn (J.Y.)

**Keywords:** sEMG, muscle synergy, non-negative matrix factorization, time-frequency analysis, wavelet packet

## Abstract

Muscle synergy analysis is a kind of modularized decomposition of muscles during exercise controlled by the central nervous system (CNS). It can not only extract the synergistic muscles in exercise, but also obtain the activation states of muscles to reflect the coordination and control relationship between muscles. However, previous studies have mainly focused on the time-domain synergy without considering the frequency-specific characteristics within synergy structures. Therefore, this study proposes a novel method, named time-frequency non-negative matrix factorization (TF-NMF), to explore the time-varying regularity of muscle synergy characteristics of multi-channel surface electromyogram (sEMG) signals at different frequency bands. In this method, the wavelet packet transform (WPT) is used to transform the time-scale signals into time-frequency dimension. Then, the NMF method is calculated in each time-frequency window to extract the synergy modules. Finally, this method is used to analyze the sEMG signals recorded from 8 muscles during the conversion between wrist flexion (WF stage) and wrist extension (WE stage) movements in 12 healthy people. The experimental results show that the number of synergy modules in wrist flexion transmission to wrist extension (Motion Conversion, MC stage) is more than that in the WF stage and WE stage. Furthermore, the number of flexor and extensor muscle synergies in the frequency band of 0–125 Hz during the MC stage is more than that in the frequency band of 125–250 Hz. Further analysis shows that the flexion muscle synergies mostly exist in the frequency band of 140.625–156.25 Hz during the WF stage, and the extension muscle synergies appear in the frequency band of 125–156.25 Hz during the WE stage. These results can help to better understand the time-frequency features of muscle synergy, and expand study perspective related to motor control in nervous system.

## 1. Introduction

Human movement is a complex process in the view of the generation, execution and completion, involving multiple levels of coordination [1]. The central nervous system (CNS) is the major part of the nervous system, mainly composed of the brain and spinal cord, which activates a series of combined muscles to achieve various behavioral goals. In this process, the combined muscles have a regular expression, named as “muscle synergy”. It is generally accepted that muscle synergy is a neural coordination structure that alleviates the computational burden associated with controlling movement and posture and flexibly controls the coordinated contraction of skeletal muscles to successfully perform various daily behavior [2,3]. Exploration on the muscle synergy provides insight into the control mechanism and pattern.

Over the past few years, many researchers have demonstrated the existence of muscle synergy during movement [4,5]. Neptune et al. [6] have found that the muscle synergy corresponds to the kinematics function from the surface electromyogram (sEMG) analysis during gait, and speculated that the muscle synergy is one of the most efficient ways for the CNS to generate complex motor behaviors. There has been some debate as to whether the origin of this muscular synergy is neurogenic.

Although some specific species, such as frogs [7], rats [8] and cats [9], as well as humans [10], have provided direct physiological evidence for the neural origin of muscle synergy theory, some scholars disagree this opinion [11]. Muscle synergy can be utilized as an indicator to determine muscle coordination [12]. Xiang Chen et al. [13] have also found that muscle synergy can be used to assess upper limb motor function of children with cerebral palsy [14]. Numerous studies have applied this method to analyze the muscle activation during pedaling [15], walking [16], knee movement [17] and elbow movement [13], and so on. Muscle synergy can be used to analyze muscle coordination adaptability [18].

Furthermore, muscle synergy has also been applied to assess motor function impaired in various diseases, such as stroke, motor cortical damage, cerebral palsy, Parkinson’s disease, etc. The possibility of cortical co-motility abnormality and recovery level in stroke patients can be evaluated by related indexes based on muscle co-motility [19]. All studies have shown that the muscle synergy patterns would change with function, resulting in abnormal movement. The above research was conducted mainly based on the non-negative matrix factorization (NMF) [20], which is an unsupervised learning algorithm similar to principal component analysis (PCA) [21], independent component analysis (ICA), etc. [22].

Compared with other decomposition algorithms, NMF can decompose the data matrix into the target form under certain constraints. Furthermore, all elements in the decomposition matrix have non-negative characteristics, which make the decomposition form and result have a better interpretation of muscle activation. In view of the large amount of data of multi-channel sEMG signals, this method can simplify the data form and realize simple and accurate extraction of target signals, so it is more suitable for collaborative analysis. Applying the NMF method to the analysis of upper extremity sEMG signals can effectively analyze the synergistic structure of muscles and explore the regularity of motor control from the perspective of nerve control.

Because of the multi-scale characteristics of sEMG, it is relatively one-sided to analyze the characteristics of intramuscular synergy from the perspective of single time domain. sEMG can provide both time and frequency domain activation information for muscle activation during exercise. Although the NMF method can provide insight into the overall muscular activity in single-time scale by using this method [23,24], it has limitations in exploring the time-frequency characteristics of synergy structure. Muscle synergy can present simultaneous activation information of muscle activation in time and frequency domain. Therefore, it is of significance to explore muscle synergy characteristics and motion control mechanisms in a time-frequency domain.

Decomposing non-stationary signals in the time-frequency domain based on wavelet transform [25] has the advantage of effectively extracting low-frequency information. Julien Frere et al. has used a wavelet-non-negative matrix decomposition method to extract gymnast upper limb muscle synergy and the way to raise the mutual activation of motor units (Motion Unit, MU) [26]. However, only the low-frequency signals are decomposed by wavelet transform and the frequency resolution decreases with the increase of frequency. Therefore, to remedy this deficiency, recent studies have shown that wavelets can decompose non-stationary signals better than Fourier transforms in both time and frequency domains. In particular, the wavelet packet transform (WPT) is widely used due to its high resolution at high and low frequencies [27]. The resolution of the wavelet packet signal in the high frequency part is higher than that of the binary wavelet. Combined with the non-linear and time-varying characteristics of the sEMG signals, the wavelet packet transform can be introduced to extract the time-frequency joint information of the sEMG signals [28]. Since this method decomposes the signal bandwidth through a series of different center frequency but the same filter, some signal features can be extracted from the selected frequency band. The interference features are effectively compressed, and the discernible features are enhanced [29].

Numerous studies have found that it is an effective tool to describe the time-frequency characteristics of sEMG signals under supination, pronation, flexion and extension movements [30]. Therefore, a decomposition of the time-domain sEMG signals into different time-frequency scales can be introduced. However, there is no similar analysis on time-frequency synergistic analysis. Above all, it is necessary to combine WPT and NMF to analyze the time-frequency synergy structures. To study the time-frequency synergy structures of the wrist flexion (WF)-wrist extension (WE) dynamic conversion process, this study proposes the time-frequency NMF method to analyze the sEMG signals recorded from 8 muscles in 12 healthy individuals. Firstly, the WPT method is used to convert the time domain signals into the time-frequency domain. Then, NMF is used to analyze the synergistic characteristics of the time-frequency domain sEMG signals. Finally, the results are quantitatively analyzed by using statistical analysis. The experimental results show that the combination of the two methods can achieve a synergistic analysis of the characteristic bands of the sEMG signals. It of significance to further understand the motion control mechanism of the nervous system in the future research to extract the muscle synergies in the specific frequency band during a specific stage.

## 2. Materials and Methods

### 2.1. Subjects

12 healthy subjects (9 males, 4 females, mean age: 23 ± 2 years) were recruited in this experiment. They had no history of physical or neurological disorders. Before the experiment, all subjects completed the Oldfield questionnaire [31] and signed an informed consent. The experiment was approved by the Ethical Review Board, and complied with the Declaration of Helsinki. To avoid fatigue, they did not experience strenuous exercise before experiment. All subjects had no similar previous experimental experiences.

### 2.2. Data Recording and Experiment Paradigm

#### 2.2.1. Experimental Paradigm

During the experiment, subjects sat in front of an experimental computer. The upper limb was supported by a bracket, which can promote the forearm to reach a horizontal position. Subjects maintained the elbow joint at 90° and prevented the shaking shoulder from affecting the experiment. The upper limb bracket had no auxiliary or resistance function on the movement of the forearm joint and the wrist (Figure 1a). Then, all subjects followed the object image on the screen to complete the wrist movement, and the signal acquisition interface showed the signal acquisition condition in real time, as shown in Figure 1a. Figure 1c shows the experimental diagram. Before the experiment, to keep the force of each wrist movement constant, the hand position of each subject was recorded when the wrist reached the maximum voluntary contraction force under the WF and WE movements. To ensure that the subjects can complete the corresponding action with the maximum voluntary contraction force. Each subject performed 10 trials with a 30 s rest period between each trial to avoid fatigue. In this study, each trial lasts 8 s and consists of a 2 s Transition stage (from relax to WF), a 2 s WF stage (maintain the WF), a 2 s motion conversion (MC) stage (from WF to WE) and a 2 s WE stage (maintain the WE).

#### 2.2.2. Data Recording and Processing

In this study, 8 superficial muscles are chosen (as shown in Figure 1b, biceps brachii (BB), brachioradialis (B), flexor carpi radialis (FCR), palmaris longus (PL), extensor carpi radialis (ECR), extensor digitorum (ED), extensor carpi ulnaris (ECU) and flexor digitorum superficialis (FDS)) as research objects according to previous studies [32], which are considered to be the main muscle groups involved in the wrist movement and conformed with research purpose. The sensor electrode of the acquisition system is very small, and its enclosure dimension is 27 × 37 × 15 mm. The sensor carries four electrode patches that can be affixed to the skin surface of the target muscle to collect sEMG signals and the contact size of each electrode patch is 5 × 1 mm. According to anatomical knowledge, through several specific actions, the corresponding muscle positions can be found [33]. The electrode is attached to the middle position of the corresponding muscle along the direction of the muscle fiber and the sEMG signals are collected and recorded. sEMG signals are acquired with the Trigno™ sEMG System produced by DELSYS Inc. (Natick, MA, USA), which is a wireless transmission physiological monitoring equipment [34]. It can collect sEMG signals bandwidth 20–450 Hz, at a sampling rate of 2000 samples/s. According to previous studies, the available energy of the sEMG signals is limited within the frequency range of 0–500 Hz, and the dominant energy is within the range of 0–250 Hz. Therefore, the frequency band range studied in this study is 0–250 Hz. Before the experiment, the skin was cleaned with 75% medical alcohol, and the electrodes were then placed along the muscle fibers according to the anatomical structure. The electrode positions are shown in Figure 1b. For the subsequent analysis, the raw sEMG signals need to be preprocessed. First, we deleted the signals with severe hand shaking or delayed response. Second, the main interference sources in sEMG signals data are also the power signal of 50 Hz and the internal electronic noise interference of instruments. According to these problems, this paper uses an adaptive notch filter to removes power signal of 50 Hz. Then, all sEMG signals were downsampled to 500 Hz. Finally, the signals were filtered with a 250 Hz lowpass filter and full-wave rectified to obtain the envelope sEMG signals for further analysis.

### 2.3. Derivation of Non-Negative Matrix Factorization in the Time-Frequency Domain

#### 2.3.1. Non-Negative Matrix Factorization

Numerous studies have shown that NMF method can be used to analyze the muscle synergy among multi-channel sEMG signals. In this method, the muscle activation model represents the sEMG signals, which can be decomposed into two matrices.

One of them is the muscle synergy vector matrix W and the other is the time-varying coefficient C. The decomposition model can be expressed as the following formula
(1)$$V_{mn}\approx {\left(WC\right)}_{mn}=\sum_{i=1}^{k}W_{mi}C_{in}={V^{\prime }}_{mn}$$
where $V_{mn}$ represents m channels sEMG signals with n sampling points, k represents the number of synergy modules and W represents the muscle synergy vector matrix, which reflects the activation weight of each muscle in the i-th synergy modules. In this study, the W value is normalized to (0–1). When the weight values of these muscles are above 0.5, they are considered as synergistic muscles. C is the time-varying coefficient, which represents the contribution of the i-th synergy matrices to the movement at time t. ${V^{\prime }}_{mn}$ represents the reconstructed sEMG signals.

To confirm the number of synergy modules k, 1≤k≤m. The variability accounted for (*VAF*) is introduced [35]. It is defined as follows
(2)$$VAF=1-\frac{RSS}{TSS}=1-\frac{\sum {\left(V_{mn}-{V^{\prime }}_{mn}\right)}^{2}}{\sum {\left(V_{mn}\right)}^{2}}$$
where RSS is the residual sum of squares and TSS is the total sum of squares.

Clark et al. [36] have proposed that there will be a superposition of many modules if the number of k is small. For a lower value of *VAF*, the decomposition model covers part of the useful information of the original signals. As the number (k)increases, the *VAF* value gradually increases.

k value should be chosen based on that it can represent at least 95% of the *VAF*, and the increase in *VAF* is less than 1% when extracting k+1 synergies [37]. Only then can it retain the details of the signals. When the k value is slightly larger, it means that the movement is complex; on the contrary, it means that the movement is relatively simple.

#### 2.3.2. Time-Frequency Non-Negative Matrix Factorization

This study proposes a method combining wavelet packet transform (WPT) and non-negative matrix factorization (NMF) to explore the time-varying law of muscle synergy characteristics of multi-channel sEMG signals at different frequency bands. Firstly, the sEMG signals are decomposed based on the WPT method to obtain the time-frequency domain information of the signal in multi-dimension. Then, the data is divided into different stages in time domain by adding a time window. The sEMG signals in a specific frequency band are further obtained. Finally, the NMF method is used to analyze the characteristics of muscle synergy in the specific frequency band during specific stage. Synergy analysis of the characteristic frequency bands of sEMG signals in different motion stages can be achieved.

Here, the multi-channel sEMG data is defined as matrix $V=\left\{x_{1},x_{2},\ldots ,x_{m}\right\}$, where the vector $x_{i}$ represents the i-th-channel sEMG data with length L and the m represents the channel number of the sEMG data. First, the WP method is introduced to decompose the vector $x_{i}$ into j-th resolution levels, respectively; $2^{j}$ equally-wide subspaces in frequency domain can be obtained correspondingly. The sub-signal of $x_{i}$ at the *N*th subspace on the j-th level is defined as $x_{i,j}^{N}$, which can be reconstructed by
(3)$$x_{i,j}^{N}\left(t\right)=\sum_{\tau }XD_{\tau }^{i,j,N}X_{i,\varphi_{N,\tau }}\left(t\right)$$
where i,j,τ∈Z, $N=1,2,\cdot \cdot \cdot ,2^{j}$. $XD_{\tau }^{i,j,N}$ is the wavelet packet coefficient at the *N*th subspace on the j-th level; $X_{i,\varphi_{N,\tau }}\left(t\right)$ is the wavelet function for $x_{i}$. If the highest frequency of the sEMG signals $x_{i}$ is $f_{H}$, the width of frequency band at the N-th subspace will be between $\frac{f_{H}}{2^{j}}\left(N-1\right)$ and
$\frac{f_{H}}{2^{j}}N$. Then, the sub-signals $x_{i,j}^{N}$ between $\frac{f_{H}}{2^{j}}\left(N-1\right)$ and $\frac{f_{H}}{2^{j}}N$ are divided into l segments with equivalent length by multiplying an appropriate time window w(t) and the $w\left(t\right)$ width is equal to $\frac{L}{l}$. Thus, the time-frequency sub-signals $x_{i,j,k}^{N}$ at the range $\frac{f_{H}}{2^{j}}\left(N-1\right)$~$\frac{f_{H}}{2^{j}}N$ in the interval $\frac{L}{l}\left(k-1\right)$~$\frac{L}{l}k$ can be obtained. Similarly, other vectors in the matrix $V=\left\{x_{1},x_{2},\ldots ,x_{m}\right\}$ can be analyzed as described above.

Then, the sub-signals of V at the N-th subspace on the j-th level in the interval $\frac{L}{l}\left(k-1\right)$~$\frac{L}{l}k$ can be obtained and the sub-signals under the same time-frequency window is combine with a m×l matrix $V_{j,k}^{N}=\left\{x_{1,j,k}^{N},x_{2,j,k}^{N},\ldots ,x_{i,j,k}^{N},....x_{i,m,k}^{N}\right\}$. Finally, the $V_{j,k}^{N}$ is decomposed into two matrices $W_{j,k}^{N}$ and $C_{j,k}^{N}$. It can be expressed as the following formula
(4)$${\left(V_{j}^{N}\right)}_{ml}\approx {\left(W_{j}^{N}C_{j}^{N}\right)}_{ml}=\sum_{p=1}^{\psi }{\left(W_{j}^{N}\right)}_{mp}{\left(C_{j}^{N}\right)}_{pl}={\left(V_{j}^{N}\right)}_{ml}^{\prime }$$
where ${\left(V_{j}^{N}\right)}_{ml}$ represents the matrix of m channels $V_{j,k}^{N}$, l is the number of sampling points and ψ represents the number of synergy modules. $W_{j,k}^{N}$ is the muscle synergy vector matrix, which reflects the activation weight of each muscle in the p-th synergy modules. $C_{j,k}^{N}$ is the time-varying coefficient and represents the contribution of the p-th synergy matrices to the movement at time t. ${\left(V_{j}^{N}\right)}_{ml}^{\prime }$ is the reconstructed sEMG signals. The muscle synergy matrix W and the time-varying coefficient C across all time-frequency domain can be obtained.
(5)$$\begin{array}{cc} W=\left\{\begin{array}{l} W_{1,1}^{N},\;W_{1,2}^{N},\;\cdot \cdot \cdot ,\;W_{1,k}^{N},\;\cdot \cdot \cdot ,\;\;W_{1,\frac{L}{l}}^{N}\\ W_{2,1}^{N},\;W_{2,2}^{N},\;\cdot \cdot \cdot ,\;W_{2,k}^{N},\;\cdot \cdot \cdot ,\;W_{2,\frac{L}{l}}^{N}\\ \cdot \cdot \cdot \;\;\;\;\;\;\;\;\;\cdot \cdot \cdot \;\;\;\;\;\;\;\;\;\;\;\;\;\cdot \cdot \cdot \;\\ W_{j,1}^{N},W_{j,2}^{N},\cdot \cdot \cdot ,\;W_{j,k}^{N},\;\cdot \cdot \cdot ,\;W_{j,\frac{L}{l}}^{N}\\ \cdot \cdot \cdot \;\;\;\;\;\;\;\;\;\cdot \cdot \cdot \;\;\;\;\;\;\;\;\;\;\;\;\;\cdot \cdot \cdot \;\\ W_{N,1}^{N},W_{N,2}^{N},\cdot \cdot \cdot ,W_{N,k}^{N},\cdot \cdot \cdot ,W_{N,\frac{L}{l}}^{N} \end{array}\right\} & ,\;C=\left\{\begin{array}{l} C_{1,1}^{N},\;C_{1,2}^{N},\;\cdot \cdot \cdot ,\;C_{1,k}^{N},\;\cdot \cdot \cdot ,\;\;C_{1,\frac{L}{l}}^{N}\\ C_{2,1}^{N},\;C_{2,2}^{N},\;\cdot \cdot \cdot ,\;C_{2,k}^{N},\;\cdot \cdot \cdot ,\;C_{2,\frac{L}{l}}^{N}\\ \cdot \cdot \cdot \;\;\;\;\;\;\;\;\;\cdot \cdot \cdot \;\;\;\;\;\;\;\;\;\;\;\;\;\cdot \cdot \cdot \;\\ C_{j,1}^{N},C_{j,2}^{N},\cdot \cdot \cdot ,\;C_{j,k}^{N},\;\cdot \cdot \cdot ,\;C_{j,\frac{L}{l}}^{N}\\ \cdot \cdot \cdot \;\;\;\;\;\;\;\;\;\cdot \cdot \cdot \;\;\;\;\;\;\;\;\;\;\;\;\;\cdot \cdot \cdot \;\\ C_{N,1}^{N},C_{N,2}^{N},\cdot \cdot \cdot ,C_{N,k}^{N},\cdot \cdot \cdot ,C_{N,\frac{L}{l}}^{N} \end{array}\right\} \end{array}$$

In this study, the $W_{j,k}^{N}$ value is normalized to (0–1). When the weight values of these muscles are above 0.5, they are considered as synergistic muscles. The *VAF* method is introduced to confirm the number of synergy modules k in each time-frequency component.

To quantitatively describe the changes about the number of synergy modules among WF, MC and WE three stages during this movement, the within-subject factors are subjected to one-way repeated measures analysis of variance (ANOVA), and the total number of the synergy modules under different the movement stage is regarded as the dependent variable. In this study, the Greenhouse-Geisser is used to correct the degree of freedom, and an alpha of p<0.05 is of significance. SPSS 19.0 for windows (SPSS Inc., Chicago, IL, USA) is used for the statistical result analysis.

## 3. Results

### 3.1. Time-Frequency Analysis for Each Muscle

To obtain the time-frequency domain characteristics from the sEMG data in 8 muscles collected based on the experimental process described above, the WPT method is used to decompose the mean sEMG signals of 10 trials from the same subject who was selected randomly. Considering that the movement task is highly correlated with the energy of the sEMG signals, the wavelet packet coefficient is used as a representation of the time-frequency features.

Since the sEMG signals have clear physical significance during specific frequency, the appropriate WPT resolution levels are chosen to clearly obtain the characteristics of the signals. Numerous studies on sEMG signals analysis with the WPT have mainly chosen three to five resolution levels, and considered that the energy under four resolution levels is highly different compared to the other resolution levels [19]. Therefore, in order to refine the frequency characteristics of the signals, in this study, four resolution levels are chosen and the whole frequency band (0–250 Hz) is divided into 16 frequency bands accordingly. The width of the window function is equal to the number of samples per second. Figure 2 shows the time-frequency distribution of the sEMG signals in ED, FCR, ECU, FDS, ECR, PL, B and BB muscles for one subject (S1), respectively. It can be seen that the amplitude of all muscles presents a regular distribution during the entire time window. As Figure 2 shows, the FCR, FDS, PL, BB and ECU muscles show a higher amplitude value at the range of 20–230 Hz during the transition (1~2 s) and WF (2~4 s) stages and the amplitude in FDS muscle is higher than that in other muscles. However, the ED, ECU, ECR and ED muscles show a higher value from 30 Hz to 220 Hz during the MC (5~6 s) and WE (6~8 s) stages, and the amplitude of ECU muscle is also higher than that of other muscles. FCR, FDS and PL muscles, which are WF agonists, and ED, ECU and ECR muscles, which are WE antagonists, are activated before the onset of the movement. Furthermore, the amplitude of the sEMG signals in ECU muscle is also high in both WF and WE stages, but it was higher in the WE stage than in WF stage. However, the BB muscle mainly shows high amplitude value in 30–100 Hz during the whole movement. Different participants have similar pairs of synergistic muscles, but they present in different synergistic modules.

### 3.2. The Number of Synergistic Modules in Different Time-Frequency Data Segments

To explore the time-frequency synergy characteristics in the above sEMG data, the sEMG data in 16 frequency bands is divided into 8 segments of equal length with a time window width of samples 500 sampling points per second. Then, the muscle synergies are extracted from the 128 data segments. Taking into account the differences between data segments, the value of *VAF* is calculated according to the above principle before analyzing the muscle synergy. Figure 3 shows the grand average k value of 10 trials for this subject under different time-frequency domain. Here, the 1–16 serial numbers correspond to 0–15.625 Hz, 15.625–31.25 Hz, 31.25–46.875 Hz, 46.875–62.5 Hz, 62.5–78.125 Hz, 78.125–93.75 Hz, 93.75–109.375 Hz, 109.375–125 Hz, 125–140.625 Hz, 140.625–156.25 Hz, 156.25–171.875 Hz, 171.875–187.5 Hz, 187.5–203.125 Hz, 203.125–218.75 Hz, 218.75–234.375 Hz and 234.375–250 Hz, respectively. It can be seen that the number of red squares in MC stage is more than that in the WF stage and WE stage. Figure 3 shows a difference between the 1st and 2nd second in the motion conversion process (MC stages) of the wrist movement. The number of synergistic modules at the 2nd second exceeds the number of synergistic modules at the 1st second during the motion maintenance process (WE and WF stages).

To investigate the differences among the three stages in all subjects, the one-way ANOVA is applied in this study. Based on statistical analysis, the total number of synergy modules for all frequency bands in each state is calculated respectively. As shown in Figure 4, it can be seen that there exists obvious statistical differences between WF stage and MC stage (F(1,24) = 22, *p* = 0.044) and there also exists obvious statistical differences between WE stage and MC stage (F(1,24) = 30.7, *p* = 0.032). However, there are no statistical differences between the WF stage and WE stage (F(1,24) = 1.261, *p* = 0.442). Therefore, the sum of the k value in MC stage is higher than that in the WF stage and WE stage.

### 3.3. Muscles Synergy Analysis within Time-Frequency Domain

In this study, in order to compare and analyze the difference between NMF and TF-NMF in muscle synergy analysis, we first decomposed the average sEMG signals from 10 trials of the same randomly selected subject based on NMF decomposition. According to Section 2.3.1, we select the number of synergistic modules as 3 to meet the experimental requirements.

After that, Figure 5 showed the analysis results of muscle synergy estimated by NMF. We found that the BB muscles are activated in module $W^{\left(3\right)}$, and play a role in the whole process of wrist movement. There are ten pairs of synergy muscles in the module $W^{\left(1\right)}$: (ED-ECU, ED-ECR, ED-B, ED-BB, ECU-ECR, ECU-B, ECU-BB, ECR-B, ECR-BB and B-BB) and six pairs of synergy muscles in the module $W^{\left(2\right)}$: (FCR-PL, FCR-FDS, FCR-BB, PL-BB, PL-FDS and FDS-BB). We found that coefficients $C^{\left(1\right)}$ and $C^{\left(2\right)}$ have peak values and $C^{\left(1\right)}$ peaks at the WE stage and $C^{\left(2\right)}$ peaks at the WF stage. Therefore, it can be concluded that the muscles activated in module $W^{\left(1\right)}$ are extensors, and those activated in module $W^{\left(2\right)}$ are flexors. The CNS activates these synergistic muscle pairs to support the WF stage and WE stage. BB muscle is the supporting muscle, and plays an important role in the whole process of wrist movement.

To further explore the law of muscle activation in different frequency bands and in different movement stages, TF-NMF decomposition was used to *decompose* the same sEMG signals as NMF decomposition. Figure 6 shows the result of the synergistic muscles in the 0–125 Hz frequency band, and each column in Figure 6 represents the muscle synergy vector matrixes W. To make the result obvious, the weight value is kept above 0.5. It can be seen that in almost all bands the flexor muscles and extensor muscles are activated in one of the modules and extensor muscles are activated in one of the modules W during MC stage (the 6th second), which result in synergy muscle pairs (FCR-PL, FCR-FDS or PL-FDS) and (ECU-ECR, ECU-B or ECR-B), except in the 0–15.625 Hz and 46.875–62.5 Hz bands, where only the extensor muscle acts in one of the modules W. During the WF stage (the 4th second), the flexor muscles are activated in only one of the modules W during the frequency band of 0–15.625 Hz and 46.875–62.5 Hz. In the WE stage (the 8th second), the extensor muscles are activated in almost all frequency bands except the 31.25–46.875 Hz and 46.875–62.5 Hz frequency bands, which don’t have synergistic extensor muscles. Furthermore, as shown in Figure 6, the weight of BB muscle is higher in one of the modules W within each frequency band during each time stage.

Figure 7 shows the result in the 125–250 Hz band compared to the result in the 0–125 Hz band. The synergy muscle pairs (ECU-ECR, ECU-B or ECR-B) are in one of the modules *W* in almost all bands during MC stage (the 6th second), except the synergy muscle pairs (FCR-PL, FCR-FDS or PL-FDS) and (ECU-ECR, ECU-B or ECR-B) only in the 125–140.625 Hz, 156.25–171.875 Hz and 171.875–187.5 Hz frequency bands. In the WF stage (the 4th second), there are synergistic flexor muscle pairs of FCR-FDS, FCR-PL, PL-FDS, PL-FDS in 140.625–156.25 Hz, 156.25–171.87 Hz, 171.875–187.5 Hz and 234.375–250 Hz frequency bands, respectively. The synergistic extensor muscle pairs of ED-ECR, ED-ECU, ECU-ECR are activated in all frequency bands, except the 171.875–187.5 Hz and 187.5–203.125 Hz frequency bands in the WE stage (the 8th second) where there are no synergistic extensor muscles. Similarly, the BB muscle is also activated in all frequency bands over all time periods.

To investigate the differences of synergistic muscle under all frequency bands in the three stages from all subjects, the synergy results for other subjects are summarized in Figure 8. It shows the result of the synergistic muscles during WF, WE and MC stages and the 1–16 serial numbers correspond to the above 16 frequency bands, respectively. The color shades represent the number of subjects with different muscle synergy. Figure 8a indicates the activation result of flexor muscle during WF stage and the abscissa represents synergistic muscle FCR-PL, FDS-PL, FCR-FDS, FDS-FCR-PL and no flexor muscle synergy, respectively. It can be seen that the flexors muscles of the other subjects are activated in the 10th frequency band, and the synergistic flexor is sporadically activated in other bands. Figure 8b represents the activation result of extensor muscle synergy in the WE stage. The abscissa represents ED-ECR, ED-ECU, ECR-ECU, ED-ECR-ECU and no extensor muscle synergy, respectively. It can be seen that only the extensor muscle synergy of the other subjects in the 9–10th band is activated. In other frequency bands, however, no regularity can be found in Figure 8b. The muscle activation state in the MC stage is shown in Figure 8c, and the abscissa indicates that there are extensor muscle synergy and flexor muscle synergy (ES and FS), synergy of the flexor muscle (FS), synergy of the extensor muscle (ES) and no muscle synergy, respectively. It can be seen that the number of flexor and extensor muscle synergy in the 1–8th frequency band is more than that in the 9–16th frequency band during the MC stage.

By comparing the experimental results of the NMF and TF-NMF methods, we found that the traditional NMF method could only analyze the time-space characteristics of muscle synergy without considering the rhythm characteristics. The TF-NMF method can more effectively and simultaneously analyze the time and frequency characteristics of muscle synergy.

## 4. Discussion

The significance of this study is the analysis of time-frequency muscle synergy during the wrist movement based on the TF-NMF method. The results show that the number of synergy modules in the MC stage is more than that in the WF stage and WE stage. The flexion muscle synergy mainly occurs in the frequency band of 140.625–156.25 Hz during the WF stage, and the extension muscle synergy occurs in the frequency band of 125–156.25 Hz during the WE stage. It is found that the number of flexor and extensor muscle synergy in the frequency band of 0–125 Hz is more than that in the frequency band of 125–250 Hz during the MC stage. It demonstrates the existence of correlated neural inputs for synergistic muscles during the dynamic wrist movement.

### 4.1. Change Mechanism about the Number of Synergy Modules in Muscle Synergy

Muscle synergy structure represents the co-activation pattern of muscles [38]. In this study, it is found that the total number of synergy modules in MC stage is higher than that in WF stage and WE stage. From the view of synergy muscles, it is speculated that synergistic flexors counteract the force output of the WF movement [39]. The synergistic extensors are activated to perform the WE movement in the MC stage. Thus, more modules are required to perform the motion process. Tang et al. [13] have also proposed that the differences of muscle synergy in structure represents the change of transmission instructions and pathological in the CNS. From an anatomical view, CNS needs to rapidly integrate information to cancel motion command and generate the corresponding motor commands during the dynamic motor task [40]. Therefore, the number of decomposition modules is higher in the MC stage than that in the other stage. It is found that the neural control motor pattern is relatively complex and requires more synergistic muscles activation to complete the transition process of movement [36]. It is found that the number of synergistic modules at the 2nd second exceeds the number of synergistic modules at the 1st second during the motion maintenance process (WE and WF stages), which can be explained that during the maintenance phase of exercise, CNS will increase the activity of the synergistic muscles over time in order to maintain a steady state [41]. These results show that muscle synergy as an effective analysis model can reveal the complex motor control mechanism. Tresch et al. [8] have also found that the CNS of the mammal achieves movement by controlling muscle synergy, and the findings in this study also validate the hypothesis that the CNS employs modular control.

### 4.2. Muscle Synergy Mechanisms during Different Frequency Band

This study further analyzes the differences about muscle synergy with frequency bands under different movement states, which shows that the flexion muscle synergy mainly occurs in the 140.625–156.25 Hz frequency band during the WF stage and the extension muscle synergy occurs in the 125–156.25 Hz frequency band during the WE stage. Strazza A. et al. [42] have also found that the common frequency band with the maximum information content is mainly concentrated at 70–160 Hz when the multi-channel muscle is activated, which can be explained from the energy metabolism and number of activated cells in the muscle gradually changing and causing fatigue during a long period of continuous movement. In order to overcome the fatigue and maintain a stable state, the CNS would automatically adjust the dominating mode of the moving muscles and enhance the activity of the synergistic muscles. Muscle activation in the high-frequency band corresponds to the rapid organization and transmission of information by the CNS [40]. Therefore, muscle synergy occurs in this band among all subjects.

Furthermore, compared to the muscle synergy for the frequency band of 125–250 Hz, it is found that the synergy muscle pairs (FCR-PL, FCR-FDS or PL-FDS) and (ECU-ECR, ECU-B or ECR-B) are activated in the frequency band of 0–125 Hz, except only the extensor muscle acts in one of the modules *W* during the frequency band of 0–15.625 Hz and 46.875–62.5 Hz, which can be explained that due to the motor task from WF stage to WE stage, the CNS could recruit the neural information of extensor muscles to perform the WE task in MC stage [43]. This study assumes that in this process, synergistic flexors can counteract the output of force. The minor differences in structural anatomy could cause a different reaction in muscle activity, and command transmission has different transmission speeds on different paths. It indicates that the force exerted by the FCR, PL or FDS muscles would disappear in each frequency band with different speeds when the wrist extension command is issued by CNS. The flexor muscles have different degrees of fatigue due to long-term movement, and fatigue is associated with the disappearance of autonomic contraction force [44]. Muscle synergy is also related to force maintenance, which leads to different degrees of reduction in flexor muscle synergy. Therefore, flexor muscle synergy does not exist in all frequency bands. Likewise, the synergy muscle pairs (FCR-PL, FCR-FDS or PL-FDS) are only activated in the frequency bands of 125–140.625 Hz, 156.25–171.875 Hz and 171.875–187.5 Hz. It can be seen that flexor muscle synergy is not obvious in these frequency bands, which can be explained that the energy of sEMG signals is mainly concentrated at 0–160 Hz, and much research has selected the sEMG below 150 Hz for corticomuscular coupling analysis [45], electro-cortical activity during sleep [46] and fatigue duration. However, the extensor muscles pair (ECU-ECR, ECU-B or ECR-B) is activated in all bands, which can be explained that the muscle synergy characteristics mainly relate to force production. In this process, the extensor muscles need to be recruited to perform the wrist extension movement. Previous studies have demonstrated that sustained movement leads to changes in muscle activation and frequency band [41]. Further analysis has showed that the BB muscle is also activated in all frequency bands during all time periods. It plays a supporting role in the whole movement, which is similar to the conclusion that the synergistic modules of wrist flexion and wrist extension have a shared synergy module based on BB muscle, as it can support the arm during the movement [47]. It can be concluded that the coding patterns of neural control motors used to generate various movements are different among the different time periods. In conclusion, muscle synergy can be used to divide co-acting muscles, and provide the theoretical basis for studying the motor control mechanism of the CNS.

## 5. Conclusions

In this study, the TF-NMF method is used to analyze the muscle synergy under time-frequency domain during the WF-WE movement in healthy individuals. The result that the number of synergy modules in the MC stage is more than that in the WF stage and WE stage shows that neural control motion patterns have relative complexity. The number of synergistic modules at the 2nd second exceeds the number of synergistic modules at the 1st second during the motion maintenance process (WE and WF stages). The flexion muscle synergy mainly occurs in the 140.625–156.25 Hz frequency band during the WF stage, and the extension muscle synergy occurs in the 125–156.25 Hz frequency band during the WE stage. The amount of flexor and extensor muscle synergy in the frequency band of 0–125 Hz during the MC stage is more than that in the frequency band of 125–250 Hz, which indicates that the energy of muscle movement is concentrated in this frequency band. These results demonstrate the time-frequency domain mechanisms of the synergistic modulation in the motor-control system. This study is expected to reveal the intrinsic mode of the nervous system function, and provide a corresponding theoretical basis for sports rehabilitation.

## Figures and Tables

**Figure 1 entropy-24-00707-f001:**
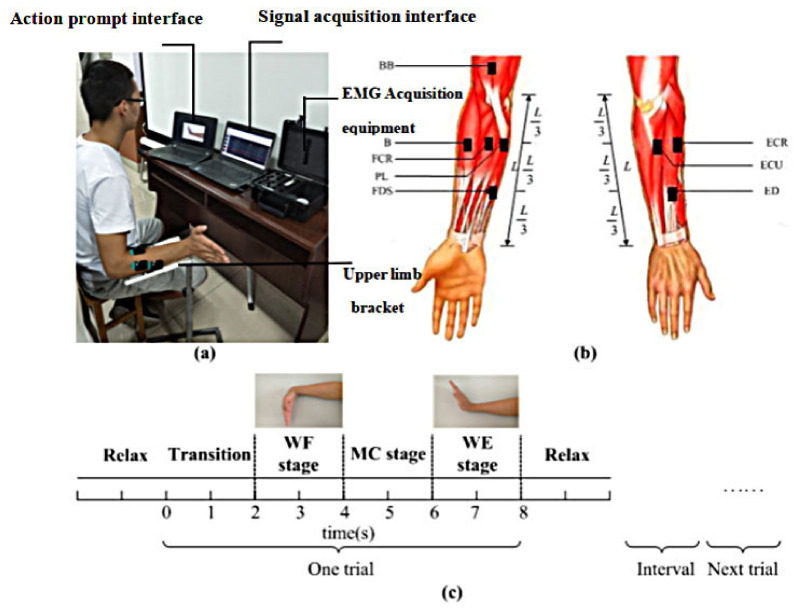
The diagram of the experiment. (**a**) Experimental setup. (**b**) The diagram of the electrode position. (**c**) The flow of the experimental task.

**Figure 2 entropy-24-00707-f002:**
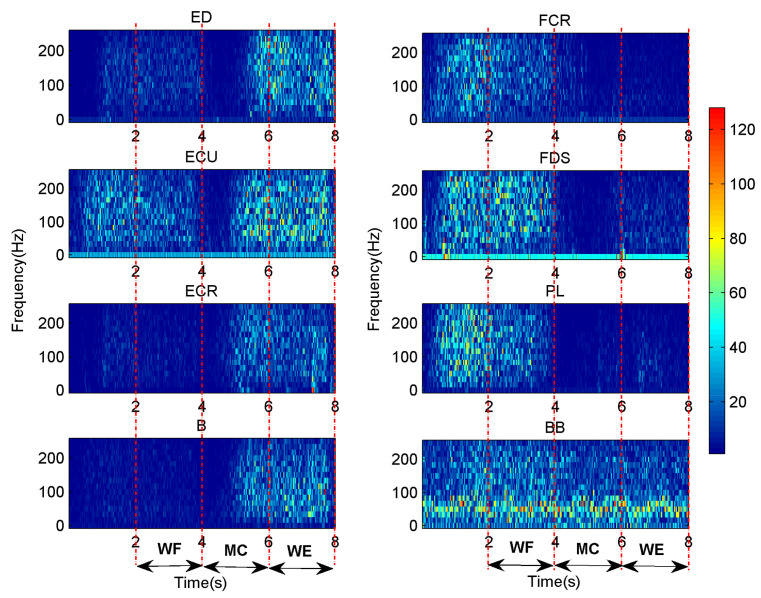
Time-frequency distributions of the sEMG signals in ED, FCR, ECU, FDS, ECR, PL, B and BB muscles for one subject (S1), respectively.

**Figure 3 entropy-24-00707-f003:**
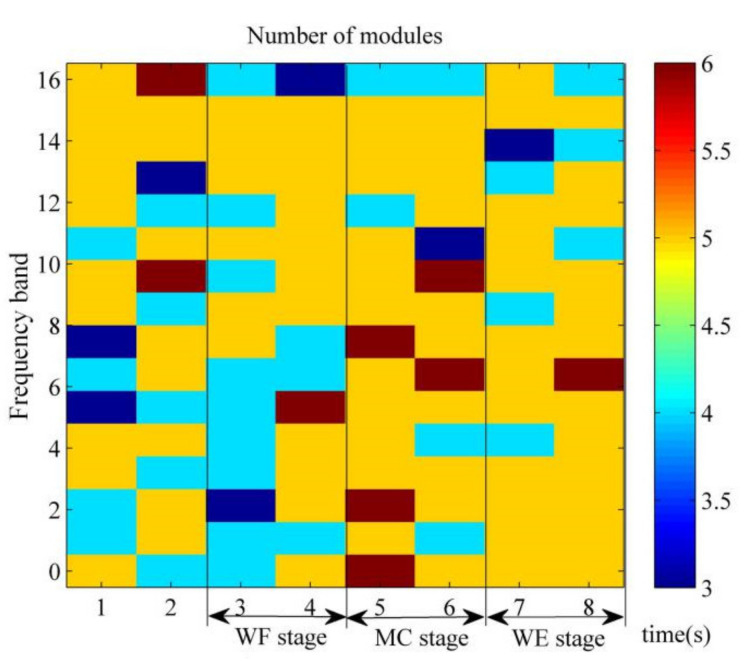
The number of synergy modules under different time-frequency domain.

**Figure 4 entropy-24-00707-f004:**
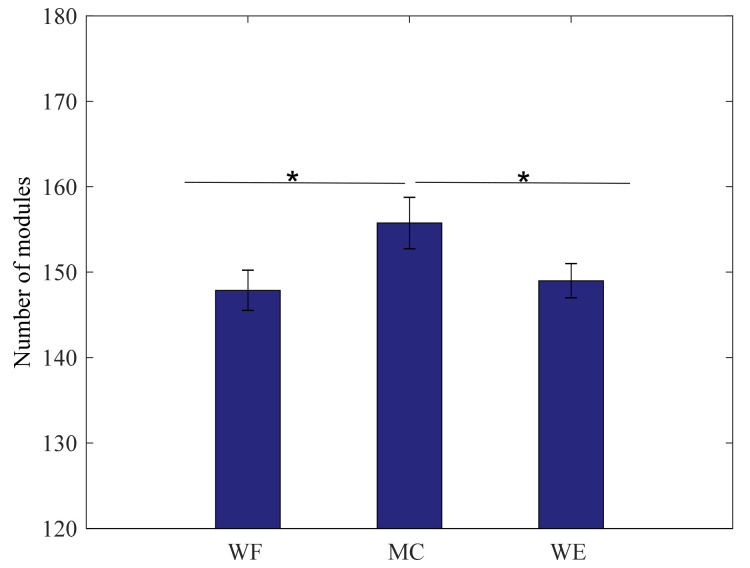
Grand averages about the number of synergy modules in the three stages. We denoted the significance with the star mark. * *p* < 0.05.

**Figure 5 entropy-24-00707-f005:**
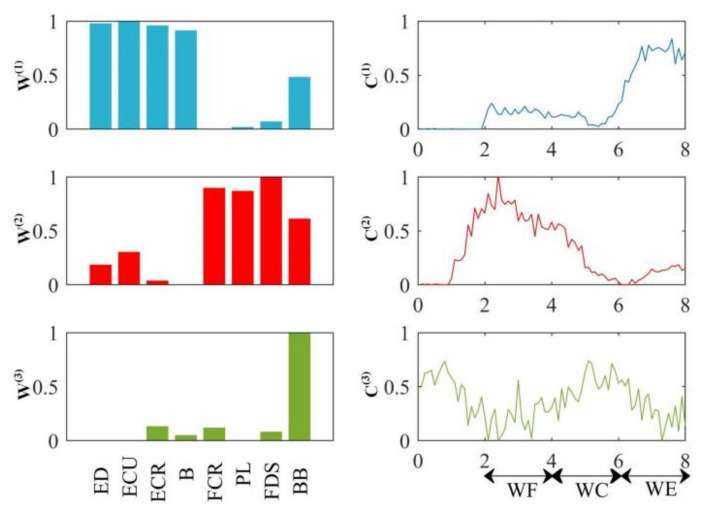
Muscle activation model based on NMF.

**Figure 6 entropy-24-00707-f006:**
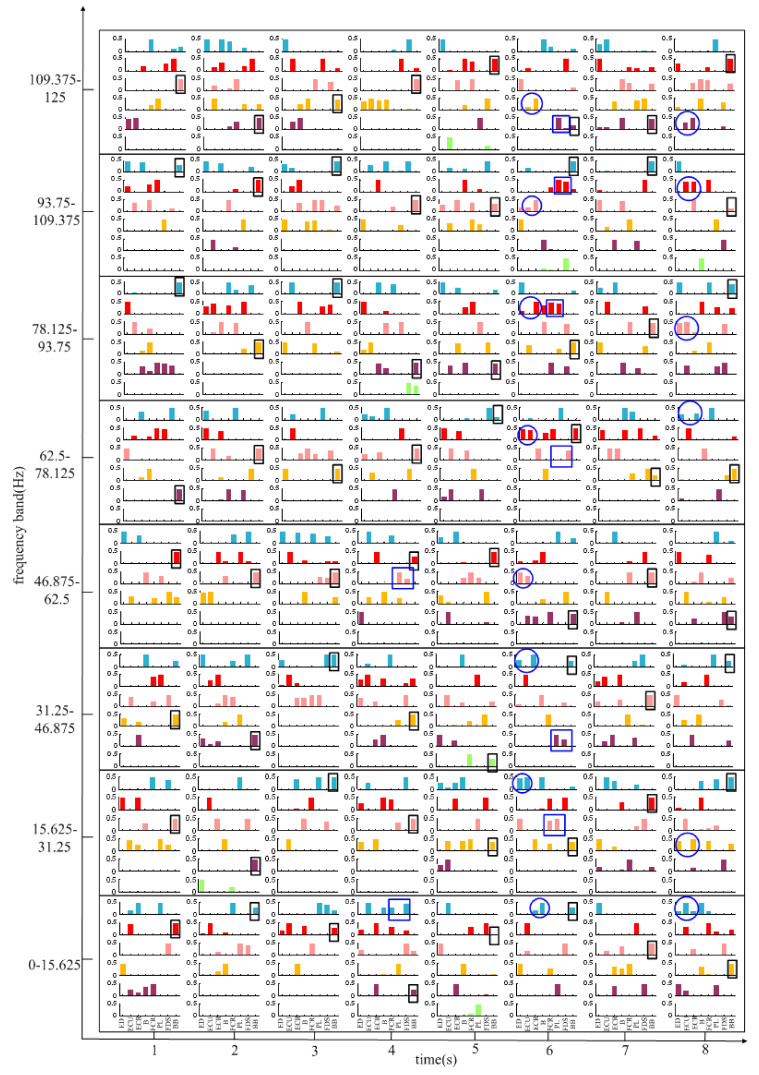
Muscle synergy vector matrixes for the frequency band of 0–125 Hz in all time periods.

**Figure 7 entropy-24-00707-f007:**
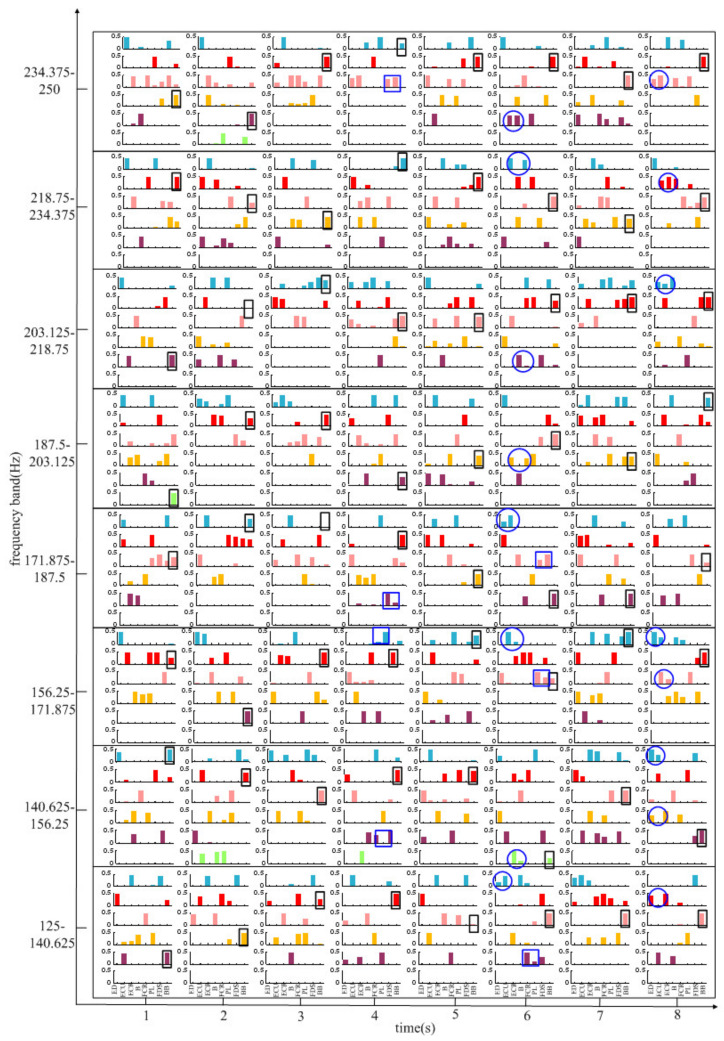
Muscle synergy vector matrixes for the frequency band of 125–250 Hz in all time periods.

**Figure 8 entropy-24-00707-f008:**
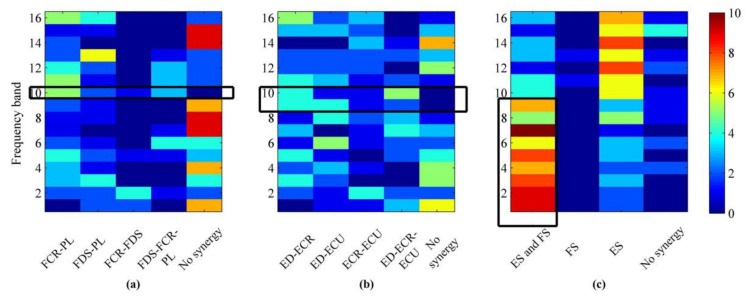
Synergistic muscle activation result of other subjects during the WF, WE and MC stages. (**a**) The distribution of the number of synergy modes in the WF stage. (**b**) The distribution of the number of synergy modes in the WE stage. (**c**) The distribution of the number of synergy modes in the MC stage. The color represents the number of people in the corresponding frequency band in this mode.

## Data Availability

Not applicable.

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
