# Peer review of "Estimation of Time-Frequency Muscle Synergy in Wrist Movements"

_entropy, 2022, doi:10.3390/e24050707_

Round 1

Reviewer 1 Report

This study proposes a novel method, named time-frequency non-negative 14 matrix factorization (TF-NMF) to explore the time-varying regularity of muscle synergy 15 characteristics of multi-channel surface electromyogram (sEMG) signals at different frequency 16 bands. The significance of this study is the analysis of time-frequency muscle synergy during the wrist movement based on the TF-NMF method. The results show that the number of synergy modules in MC stage is more than that in WF stage and WE stage. The flexion muscle synergy mainly occurs in the frequency band of 140.625-156.25 Hz during WF stage and the extension muscle synergy occurs in the frequency band of 125-156.25 Hz during WE stage. It is found that the number of flexor and extensor muscle synergy in the frequency band of 0-125 Hz is more than that in the frequency band of 125-250 Hz during the MC stage.  This topic is overall interesting. The proposed method has a novelty in the technical aspect of muscle synergy analysis. However, some revisions are needed to improve the quality of manuscript. Please see attached.

Reviewer 2 Report

General Comments

The abstract could be shorter! Perhaps present less details on the results obtained.

The title “Time-frequency estimation in muscle synergy during wrist movements” – “estimation” of what? Consider replacing “during” since it is implicit in “wrist movements”.

Overall this article presents some advancements concerning monitoring features of muscle synergy and provides some new insights into the time-frequency domain analysis.

Content

Line 35 – CNS is not yet defined.

Line 44 – reference to “Neptune et al.” without reference number, provide one.

Line 122 – Get a reference for “Oldfield questionnaire”

Lines 160-161 – The range does not seem to be right! “It can collect data in the bandwidth range of 0.5-450Hz with frequency sample at about 2000Hz.” – the frequency bands required to be observed should be clearly stated and referenced. Needs to be clearly stated in order to sustain data acquisition as well as relevance of data for this study.

Line 294 – states the time with of 500. – what are the units?

Line 298-301 – Frequency bands are presented with 3 decimal points – although sampling is performed at 2000 Hz it should be considered the significance of this 3 decimal places!

Line 392 – “Tang et al.” should be referenced.

Round 2

Reviewer 1 Report

No further revision needed.